# The Prediction of Radiation-Induced Trismus by the Apparent Diffusion Coefficient Values of Masseter Muscles before Chemoradiotherapy in Locally Advanced Nasopharyngeal Carcinomas

**DOI:** 10.3390/diagnostics14202268

**Published:** 2024-10-12

**Authors:** Umur Anil Pehlivan, Efsun Somay, Cigdem Yalcin, Erkan Topkan

**Affiliations:** 1Adana Dr. Turgut Noyan Application and Research Center, Department of Radiology, Faculty of Medicine, Baskent University, Adana 01120, Turkey; 2Department of Oral and Maxillofacial Surgery, Faculty of Dentistry, Baskent University, Ankara 06490, Turkey; efsuner@gmail.com; 3Adana Dr. Turgut Noyan Application and Research Center, Department of Radiation Oncology, Faculty of Medicine, Baskent University, Adana 01120, Turkey; docdretopkan@gmail.com

**Keywords:** apparent diffusion coefficient, masseter muscle, nasopharyngeal carcinoma, radiation-induced trismus

## Abstract

Purpose: Although the apparent diffusion coefficient (ADC) value from diffusion-weighted imaging can provide insights into various pathological processes, no studies have examined the relationship between the pre-concurrent chemoradiotherapy (CCRT) mean ADC (ADC_mean_) values of the masseter muscles and radiation-induced trismus (RIT) in locally advanced nasopharyngeal carcinoma (LA-NPC) patients. Therefore, the current research aimed to investigate the significance of pre-CCRT masseter muscle ADC_mean_ values for predicting the RIT rates in LA-NPC patients treated with definitive CCRT. Materials and Methods: The pre-CCRT ADC_mean_ values of the masseter muscles and the post-CCRT RIT rates were evaluated. A receiver operating characteristic curve analysis was employed to determine the optimal ADC_mean_ cutoff. The primary objective was to examine the relationship between the pre-CCRT masseter muscle ADC_mean_ values and the post-CCRT RIT rates. Results: Seventy-seven patients were included. The optimal ADC_mean_ cutoff value was 1381.30 × 10^−6^ mm^2^/s, which divided the patients into two groups: an ADC_mean_ < 1381.30 × 10^−6^ mm^2^/s (n = 49) versus an ADC_mean_ > 1381.30 × 10^−6^ mm^2^/s (n = 28). A masseter muscle ADC_mean_ > 1381.30 × 10^−6^ mm^2^/s was found to be associated with significantly higher RIT rates than an ADC_mean_ < 1381.30 × 10^−6^ mm^2^/s (71.42% vs. 6.12%; *p* < 0.001). The multivariate analysis results confirmed a pre-CCRT masseter muscle ADC_mean_ > 1381.30 × 10^−6^ mm^2^/s as an independent predictor of RIT. Conclusions: Our study presents the first evidence establishing a connection between elevated masseter muscle ADC_mean_ values and higher RIT rates in LA-NPC patients following CCRT. If confirmed with further research, these findings may help to categorize the risk of RIT in these patients.

## 1. Introduction

Nasopharyngeal cancer (NPC) originates in the nasopharyngeal epithelium and is notably common in East Asian people [1]. Locally advanced NPC (LA-NPC), which is the most common presenting form of NPC, is usually managed with intensity-modulated radiation therapy (IMRT) combined with concurrent chemotherapy, which is considered the standard treatment [2,3]. Although concurrent chemoradiotherapy (CCRT) with IMRT offers advantages, such as improved local tumor control, increased survival rates, and decreased acute and chronic complications, a notable proportion of patients with LA-NPC experience severe long-term side effects, such as radiation-induced trismus (RIT), which significantly impairs their quality of life [4,5].

RIT, as defined by Dijkstra and colleagues, is characterized by a maximum mouth opening (MMO) of 35 mm or less [6]. Although multiple pathophysiological mechanisms are involved, RIT leads to a limited mouth opening of various degrees depending on the severity of the insult. A reduced MMO is mainly caused by spasms and radiation-induced fibrosis that affect the masticatory muscles, primarily when the condition affects structures such as the temporomandibular joint (TMJ), related raphes, synovial fluid, and masticatory apparatus [7]. Various biological mechanisms, including radiation-induced severe local and systemic inflammation, endothelial damage, hypoxia, and fibrosis, contribute to the genesis of RIT, though the precise mechanism behind this phenomenon is yet to be fully elucidated [8]. Irrespective of the particular pathophysiological mechanisms, restricted jaw mobility resulting from RIT adversely impacts crucial activities like eating, swallowing, speaking, and maintaining oral hygiene [5]. Moreover, time-consuming, complex intubation procedures may be necessary under urgent circumstances and possibly threaten a patient’s life [9]. The incidence of RIT is linked to the dose of radiation administered to the masticatory apparatus and the magnitude of the provoked immune–inflammatory responses [10,11]. A recently reported study examined the pretreatment radiological variables of LA-NPC patients who had CCRT and found a significant link between a lower total masseter muscle (MM) volume before CCRT and the occurrence rates of RIT [12]. However, the existing literature lacks comprehensive examinations of the plausible associations among the various radiological markers in the context of RIT.

Diffusion-weighted imaging (DWI) has been well recognized for assessing different disease states by quantifying the stochastic movement of water molecules inside tissues. The apparent diffusion coefficient (ADC), obtained from DWI, offers quantitative information on tissue cellularity and microstructure, such that a higher ADC value suggests less cellularity [13,14]. DWI is an essential imaging tool for characterizing tumorous lesions, assessing the treatment response during follow-up, and identifying treatment-related changes and recurrent tumors. Typically, heightened cellularity in tumoral tissues correlates with lower ADC values [15]. Furthermore, DWI offers diagnostic information about non-tumoral changes. According to two investigations conducted by Hatakenaka et al. [16,17], the passive motions of joints in skeletal muscles may have an impact on the diffusivity of protons in tissues. Chikui et al. [18] and Shiraishi et al. [19] have conducted studies that explicitly investigated the impact of clenching on the ADC values of the MMs, and have reported an increase in ADC values during clenching. In a recent survey, Muraoka et al. [20] used the Eichner index to investigate alterations in the ADC of the masticatory muscles. Their findings revealed that individuals with less occlusal support had decreased ADC values for their masticatory muscles. Currently, no studies have examined the correlation between the ADC levels of the masticatory muscles prior to CCRT and the occurrence of RIT.

Cancer instigates mild-to-moderate local and systemic inflammation, which may worsen in metastatic disease states. This mild-to-moderate inflammatory response in the tissues may cause endothelial damage and extravasation of the tissue fluid [21], creating extracellular edema. Previously, it was demonstrated that the ADC is increased (*p* < 0.001) in cases of inflammation [22]. Consequently, we hypothesized that cancer-associated inflammation and extracellular edema would raise the mean ADC (ADC_mean_) values of the MMs and serve as a biomarker for predicting RIT rates after CCRT in LA-NPC patients. Therefore, we conducted this current retrospective investigation to determine the validity of this hypothesis for LA-NPC patients treated at our facility.

## 2. Materials and Methods

### 2.1. Ethical Considerations, Consent, and Permissions

This retrospective study was carried out at a single institution, carefully following the ethical rules set by the Institutional Clinical Research Ethics Committee and adhering to the principles of the Helsinki Declaration and its later amendments. Baskent University’s Institutional Review Board approved this research under Project No. KA 24/273. Before starting CCRT, all the participants gave their written consent for the collection of data and the publishing of the associated results, following our institutional policies and standards.

### 2.2. Patient Selection

This retrospective study was performed at Baskent University’s Dr. Turgut Noyan Application and Research Center on patients diagnosed with NPC who received CCRT between January 2010 and December 2023.

The inclusion criteria for this study were as follows: patients who were 18 years of age or older, had undergone a pretreatment neck magnetic resonance imaging (MRI) evaluation, had an Eastern Cooperative Oncology Group (ECOG) performance status of 0–1, had a body mass index (BMI) of 18.5 kg/m^2^ or higher, had histopathologically confirmed nasopharyngeal squamous cell carcinoma, and were staged as LA-NPC T1–2N1–3M0 or T3–4N0–3M0 according to the eighth edition of the *AJCC Staging System* [23]. To be eligible, the patients also had to not have a diagnosis of temporomandibular disorder (TMD) before CCRT, according to the latest criteria for TMD [24]; no history or signs of trismus during the initial oral exams; no head and neck injuries; and no disorders affecting the masticatory apparatus, such as the temporomandibular joint (TMJ).

The patients lacking preoperative DWI or who experienced technical difficulties measuring their ADC values; were missing maxillary and/or mandibular central incisors; had undergone prior TMJ surgery; had TMJ ankylosis, head and neck trauma, muscle-related pain or myofascial pain syndrome, primary tumor or lymph node invasion of the masticatory muscles, facial asymmetry or dental malocclusion, or pre-existing TMD; were ineligible based on the treatment criteria; had prior treatment for a local recurrence; or had a previous surgery, chemotherapy, and/or RT in the head and neck region with less than six months of follow-up were deemed ineligible for this study.

### 2.3. Baseline Dental Examination

RIT was characterized by an MMO of 35 mm or less, according to the criteria set by Dijkstra et al. [6]. Therabite**^®^** (Atos Medical AB, Hörby, Sweden) was used for measuring the MMOs because of its established measurement accuracy and user-friendly application [25]. During the MMO measurements, the patients were instructed to align their heads parallel to the Frankfurt horizontal plane and look straight ahead. The patients were further directed to fully open their mouths using the Therabite**^®^** motion scale to quantify the gap between the lower edge of one of the upper central incisors and the upper edge of one of the matching mandibular central incisors. We calculated the arithmetic average of three consecutive measurements per session to determine the mean MMO.

### 2.4. Imaging Acquisition

All the patients underwent neck MRI scans using a 20-channel head–neck coil on a Siemens MAGNETOM Skyra 3 T or MAGNETOM Avanto Fit 1.5 T MRI scanner (Siemens Healthineers, Erlangen, Germany). The MRI protocol for the neck included pre-contrast and post-contrast sequences as follows: axial, coronal, and sagittal T1-weighted spin-echo sequences, as well as axial and coronal T2-weighted spin-echo sequences. The DWI used single-shot spin-echo echo-planar imaging with diffusion sensitivities (*b*-values) of 0 and 1000 s/mm^2^ in three orthogonal directions (Z, Y, and X). The specific DWI parameters were set as follows: for the 3 T scanner, the time to repeat (TR) = 7100 ms, the time to echo (TE) = 62 ms, the slice thickness = 4 mm, and the field of view (FOV) = 230 mm × 230 mm; for the 1.5 T scanner, the TR = 7600 ms, the TE = 110 ms, the slice thickness = 3 mm, and the FOV = 230 mm × 230 mm. The phase encoding was anterior to posterior for both scanners, with a transverse orientation for the DWI. Additionally, we acquired post-contrast axial and sagittal fat-suppressed T1-weighted spin-echo sequences after administering a gadolinium-based contrast intravenously.

### 2.5. Post-Processing and Image Analysis

ADC maps were created using conventional pixel-based software (Syngo Via, Siemens, Erlangen, Germany). The MMs were identified using both the neck MRI sequences and the ADC maps. The ADC values were measured using a freehand region-of-interest (ROI) analysis, specifically targeting the thickest parts of the muscles on the axial plane. A head and neck radiologist with three years of expertise conducted the analysis. In addition, a second observer, who is a head and neck radiologist with seven years of expertise, performed measurements to evaluate the consistency of the interobserver ADC values for 10 patients. The ADC_mean_ values for each MM and the average of these values were recorded.

### 2.6. Treatment Protocol

As previously described elsewhere, IMRT was administered to all the LA-NPC patients [26]. Using a pretreatment co-registered CT, 18-FDG-PET-CT, and/or MRI scans of the affected primary NPC and the entire neck, each target volume was determined. The total prescribed doses for the high-, intermediate-, and low-risk planning target volumes were 70 Gy, 59.4 Gy, and 54 Gy, respectively, delivered in 33 daily fractions. The RT was combined with three cycles of concurrent chemotherapy, including cisplatin and 5-fluorouracil, on days 1, 22, and 43. Following the CCRT, the patients were recommended to complete two cycles of the same chemotherapy regimen used during the CCRT phase of their treatment as an adjuvant therapy if it was tolerable for them. Supportive care measures were provided to all patients when necessary.

### 2.7. Follow-Up Dental Examination

We conducted post-CCRT MMO measurements as outlined in the “Baseline Dental Examination” section. The MMO measurements were performed at 1, 3, 6, 9, and 12 months following the CCRT and every six months thereafter, using the same protocol to assess the RIT status (Figure 1).

### 2.8. Statistical Analysis

The primary objective of this retrospective study was to investigate the potential association between the MMs’ pre-CCRT average ADC_mean_ values and the RIT incidence rates after CCRT. A Kolmogorov–Smirnov test confirmed the normality of the distribution for the continuous variables. We used medians and ranges to represent the continuous data and frequency distributions with percentages to convey the categorical variables. The patients were compared using suitable statistical methods, such as the chi-square test. We employed a receiver operating characteristic (ROC) curve analysis to establish a pre-CCRT cutoff for the average of the MMs’ ADC_mean_ values that may significantly correlate with the post-CCRT RIT statuses. Spearman’s product–moment correlation coefficient was calculated to evaluate the interobserver’s correlation of measurements for the 10 sample patients. We conducted univariate analyses to explore the relationship between the patient, disease factors, treatment variables, and this study’s primary endpoint, namely the RIT rates. The variables that demonstrated a statistical significance in the univariate analysis were next evaluated in a multivariate Cox proportional hazards model to determine the independent predictors of RIT. The statistical tests conducted in this research were two tailed, and a significance threshold of *p* < 0.05 was used to evaluate the statistical significance.

## 3. Results

The present research assessed 77 patients diagnosed with LA-NPC, comprising 22 females (28.57%) and 55 males (71.43%). The group’s median age was 53, ranging from 25 to 76. Most patients had either T3–4 (59.74%) or N2–3 (66.23%) illness stages. Table 1 provides a summary of the clinical, radiological, and therapeutic features of the cohort.

The research group had a median follow-up time of 50.5 months, ranging from 10 to 101 months. A total of 54 cases (70.13%) did not experience RIT, while 23 patients, accounting for 29.87% of the total, experienced it. The median ADC_mean_ value for the MMs across all the patients was 1220.0 × 10^−6^ mm^2^/s, with a range of 548.78 × 10^−6^ mm^2^/s to 1758.30 × 10^−6^ mm^2^/s. The median ADC_mean_ value of the MMs was 1521.04 × 10^−6^ mm^2^/s (range: 1271.20 × 10^−6^ mm^2^/s–1758.30 × 10^−6^ mm^2^/s) in patients with RIT (Figure 2), which was significantly higher than the median ADC_mean_ value of 914.51 × 10^−6^ mm^2^/s (range: 548.78 × 10^−6^ mm^2^/s–1561.10 × 10^−6^ mm^2^/s) measured in the non-RIT group (*p* < 0.001) (Figure 3). A strong positive correlation was observed between the ADC measurements provided by the two reviewers for the MMs of 10 patients. The coefficient of determination (R^2^) was 0.913 for the ADC_mean_ values of the right MMs, 0.905 for the ADC_mean_ values of the left MMs, and 0.921 for the average ADC_mean_ values of both MMs.

The ROC analysis revealed that the ideal average ADC_mean_ cutoff value for the MMs was 1381.30 × 10^−6^ mm^2^/s for RIT (area under the curve: 94.0%; sensitivity: 87.0%; specificity: 85.2%; and Youden index: 0.722), with the RIT incidence status being the event (Figure 4). Therefore, the patients were categorized into two groups according to the average value of the MMs’ ADC_mean_: <1381.30 × 10^−6^ mm^2^/s (n = 49) versus >1381.30 × 10^−6^ mm^2^/s (n = 28). Comparisons between the two MMs’ ADC_mean_ cohorts revealed that the patients with an MMs’ ADC_mean_ > 1381.30 × 10^−6^ mm^2^/s experienced significantly higher rates of RIT than those with MMs’ ADC_mean_ < 1381.30 × 10^−6^ mm^2^/s (71.42% vs. 6.12%; Hazard ratio [HR]: 11.17; *p* < 0.001) (Figure 5). This significant difference in the occurrence of RIT between the two groups was achieved despite the nearly equal distribution of patients, diseases, and treatment characteristics between the two groups.

The univariate analysis demonstrated that in addition to a higher average ADC_mean_ value of the MMs (>1381.30 × 10^−6^ mm^2^/s), a higher mean masseter muscle dose (>37.5 Gy), a T3–4 tumor stage, and the administration of 2–3 concurrent chemotherapy cycles were the factors associated with significantly higher rates of RIT incidence after CCRT (*p* < 0.05 for each) (Table 2). The multivariate Cox regression analysis further indicated that all four factors retained their independent significant impact on elevated rates of RIT in this patient group (Table 2 and Figure 6).

## 4. Discussion

This research examined the correlation between the average ADC_mean_ values of the MMs before CCRT and the occurrence of RIT. We have shown, for the first time, that a higher pre-CCRT MM average ADC_mean_ value (>1381.30 × 10^−6^ mm^2^/s) is linked to significantly increased rates of RIT (71.43% vs. 6.12% for an average ADC_mean_ < 1381.30 × 10^−6^ mm^2^/s; HR: 11.17, *p* < 0.001) in LA-NPC patients who received definitive CCRT. Additionally, our results support the significance of well-known RIT risk factors, such as a higher mean masseter muscle dose (>37.5 Gy), a T3–4 tumor stage, and receiving two or three chemotherapy cycles concurrent with RT.

Consistent with the existing RIT literature, the current study establishes that the delivery of a radiation dose of approximately 37.5 Gy to the MMs, a T3–4 tumor stage, and the administration of 2–3 cycles of concurrent chemotherapy are all independent predictors associated with heightened RIT rates. According to a recently published meta-analysis, adding chemotherapy to RT significantly raises the risk of trismus development in head and neck cancer (HNC) patients by a factor of 2.55, and this increased risk has been associated with detrimental effects on patients’ overall quality of life [27]. The team at M.D. Anderson Cancer Center established a link between a higher T-stage and the occurrence of RIT after CCRT in patients with oropharyngeal cancer [28]. In line with this finding, our earlier study showed a strong link between a higher T-stage (T3–4) and higher RIT rates in LA-NPC patients treated with definitive CCRT [12]. These consistent findings suggest that a more voluminous or invasive tumor causes higher doses to be unavoidably delivered to the masticatory apparatus, increasing the risk for RIT. Confirming both this remark and the finding of a correlation between a mean MM dose > 37.5 Gy and elevated RIT rates in our study, Kraaijenga et al. [29] reported a robust connection between high-dose radiation to the masseter and medial pterygoid muscles and the development of RIT in patients with HNC undergoing CCRT.

The primary finding of this study is the first evidence of a strong association between a high average ADC_mean_ value of the pre-CCRT MMs and a giant rise in RIT rates. Specifically, despite the indistinguishable distribution of the confounding variables, the rate of RIT for an average ADC_mean_ value > 1381.30 × 10^−6^ mm^2^/s was about 12 times higher than that for the group with ADC_mean_ values < 1381.30 × 10^−6^ mm^2^/s (71.43% vs. 6.12%; *p* < 0.001). If further verified, these data indicate that the ADC_mean_ values have significant potential as a new radiomics-based predictor for RIT in patients with LA-NPC. Recent studies have underscored the importance of DWI for enhancing the functional evaluation capabilities of conventional MRI by allowing for the differentiation of various tissues and lesions, including tumors, infections, myopathy, fibrosis, and edemas [30,31]. While previous studies have examined the ADC metrics of masticatory and striated skeletal muscles for different pathological conditions [32,33], none have specifically explored the connection between the average ADC_mean_ values of the pre-CCRT MMs and their correlation with RIT rates in HNC patients, particularly those undergoing exclusive CCRT, including LA-NPCs. Sawada and colleagues used ADC values to quantify the muscular pain in the masticatory muscles of individuals with TMDs [33]. Their research found that individuals with TMDs had elevated ADC values for the masticatory muscles on the afflicted side. In a more recent study, Sawada and colleagues compared the ADC values of the masticatory muscles for patients with and without TMJ osteoarthritis (TMJ-OA) [34]. The results of this study indicated that patients without TMJ-OA exhibited significantly higher ADC values for the masticatory muscles than those with TMJ-OA. However, our study distinguishes itself from those conducted by Sawada and colleagues. Sawada’s studies used ADC values to differentiate between healthy and diseased states of the masticatory muscles for diagnostic purposes, rather than for predicting radiation-induced toxicity in theoretically healthy structures before CCRT, as our study did.

Although our study found a strong link between low pretreatment average ADC_mean_ values for the MMs and the occurrence of RIT, it is difficult to explain the exact mechanism behind this connection using evidence-based methods, since no further research exists on this topic. In cancer patients, there is mild-to-moderate local and systemic inflammation, which is the seventh hallmark of cancer [35]. Vascular permeability by any measure is dramatically increased in acute and chronic inflammation induced by any type of tissue injury, wound healing, or cancer [21,36]. The hyperpermeability state observed in cancer is responsible for the development of extracellular vasculogenic edema, which is associated with elevated ADC_mean_ values [37]. In their pioneering work, Ito and colleagues sought to evaluate the efficacy of DWI for detecting the kind of edema that arises after experimental trauma and trauma combined with hypotension and hypoxia [37]. Their research found that the ADCs were reduced (*p* < 0.001) in the case of cytotoxic edema and increased (*p* < 0.05) for extracellular edema. Because various degrees of cancer-induced inflammation and related vascular problems are present in almost every cancer patient and there is a vicious circle between inflammation and hypoxia, it is reasonable to postulate that a robust relationship between a higher pre-CCRT average ADC_mean_ and increased RIT rates may be a result of a causal relationship between these pathological conditions, which may render the masticatory apparatus more susceptible to RIT. Nevertheless, this statement should only be deemed a sensible assumption until preclinical and clinical research findings on this critically important topic are available. In this context, intravoxel incoherent motion (IVIM) imaging, an advanced diffusion imaging modality, could help assess the free water fraction, which can vary dramatically depending on the severity of the inflammatory processes [38]. Given that inflammation is a critical parameter in RIT development and progression, studies using IVIM imaging may provide greater clarity about the extravascular edema status of RIT patients, thereby establishing a link between a higher average ADC_mean_ and increased RIT rates.

The present study is subject to several limitations. Firstly, it employed a single-center retrospective design with a relatively limited cohort size, making the results vulnerable to unforeseen biases. Secondly, prior research has indicated potential variations in muscle ADC values during muscle contraction. Unfortunately, due to this study’s retrospective nature, information regarding the contraction status of the muscles during image acquisition was lacking. Nevertheless, given the absence of specific patient instructions during the imaging procedure, it is anticipated that any influence on the outcomes would have been uniform, positively or negatively, if there were any. Thirdly, while we established a correlation between ADC values—potentially reflective of the qualitative and quantitative cellular properties of the MMs and the extracellular component of the related region, a crucial component of the masticatory apparatus—and the occurrence of RIT, further investigation is necessary to explore similar associations within other components of the masticatory apparatus. In the current study, efforts were made to evaluate the ADC_mean_ values for the MMs and to examine the potential relationships of the other components of the masticatory apparatus with RIT. Unfortunately, we realized that the imaging technique was not suitable for this purpose. Hence, this issue should be investigated through prospectively designed studies to determine such a relationship’s existence. Fourth, the use of 1.5 T and 3 T MRI scanners may render the reliability of the ADC measurements questionable for some researchers. Although recent in vitro study findings have suggested that significant differences may occur [39], the majority of the available in vivo literature suggests that there is generally no significant difference in the magnetic field strength and the ADC values obtained with 1.5 T and 3 T MRI scanners [40,41]. While further investigation of this matter is merited, it is notable that we utilized two MRI systems produced by the same manufacturer, each of which was calibrated by the same quality assurance team. Consequently, we posit that the potential influence of employing two distinct magnetic field strengths and imaging parameters on the comprehensive outcomes is expected to be negligible, if existent at all. Consequently, considering all these limitations, it is prudent to regard this study’s findings as hypothetical unless substantiated by meticulously planned subsequent studies addressing these particular issues.

## 5. Conclusions

In conclusion, the present study’s findings suggest a strong and independent link between higher average ADC_mean_ values for the MMs of LA-NPC patients undergoing definitive CCRT and increased RIT incidence rates. If subsequent research corroborates these findings, assessing the association between the pre-CCRT DWI findings and the risk of RIT could facilitate the closer monitoring of high-risk patients and the timely implementation of preventive interventions. This strategy has the potential to significantly improve patients’ quality of life.

## Figures and Tables

**Figure 1 diagnostics-14-02268-f001:**
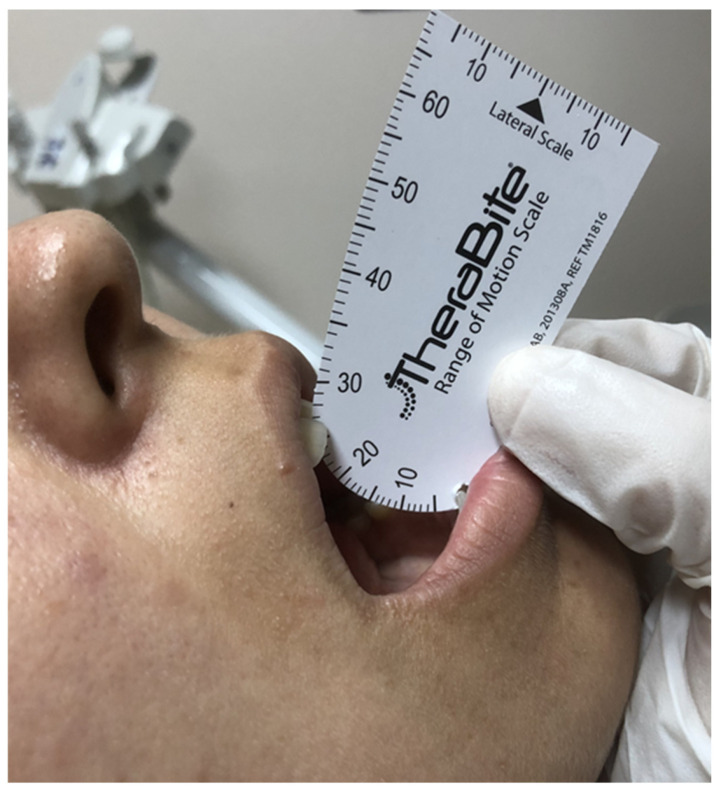
A representative examination of a patient with radiation-induced trismus (maximum mouth opening: 26 mm) at follow-up.

**Figure 2 diagnostics-14-02268-f002:**
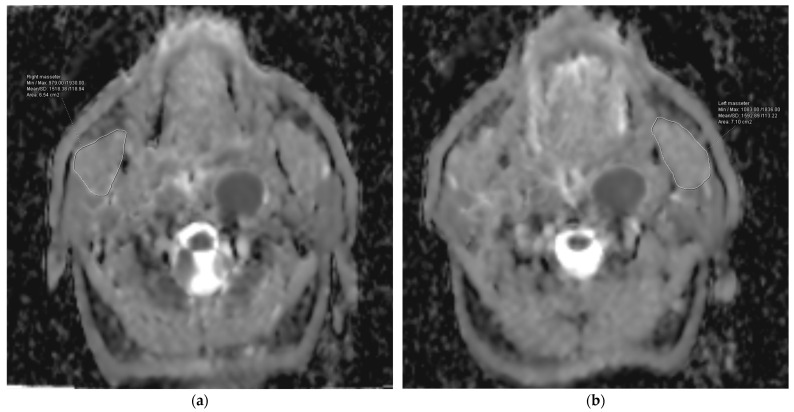
Representation of the measurement of the apparent diffusion coefficient (ADC) values for the right (**a**) and left (**b**) masseter muscles of a patient from the group with higher ADC values.

**Figure 3 diagnostics-14-02268-f003:**
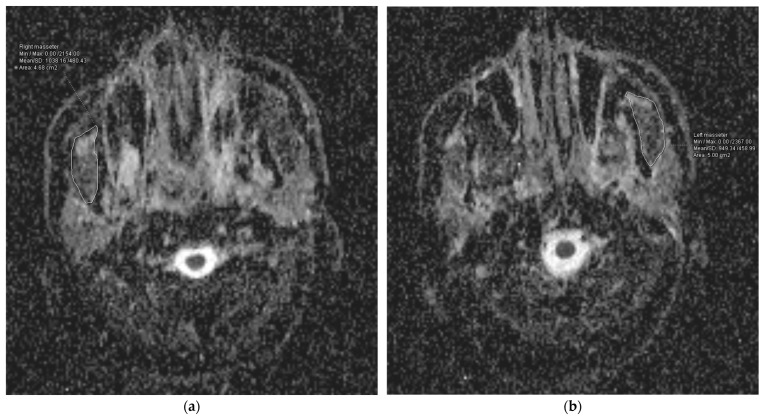
Representation of the measurement of the apparent diffusion coefficient (ADC) values for the right (**a**) and left (**b**) masseter muscles of a patient from the group with lower ADC values.

**Figure 4 diagnostics-14-02268-f004:**
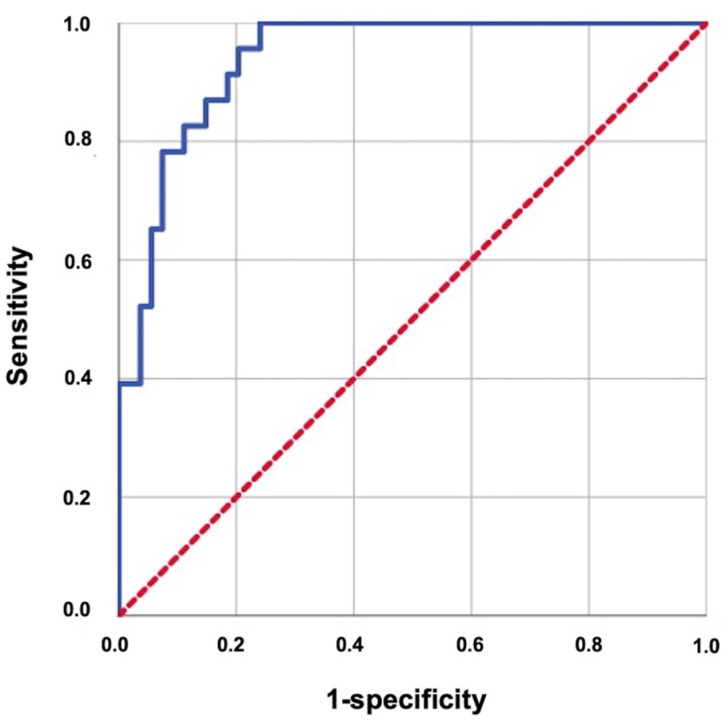
Results of the receiver operating characteristic curve analysis evaluating the relationship between the average ADC_mean_ of the masseter muscles with radiation-induced trismus (area under the curve: 94.0%; sensitivity: 87.0%; specificity: 85.2%; and Youden index: 0.722).

**Figure 5 diagnostics-14-02268-f005:**
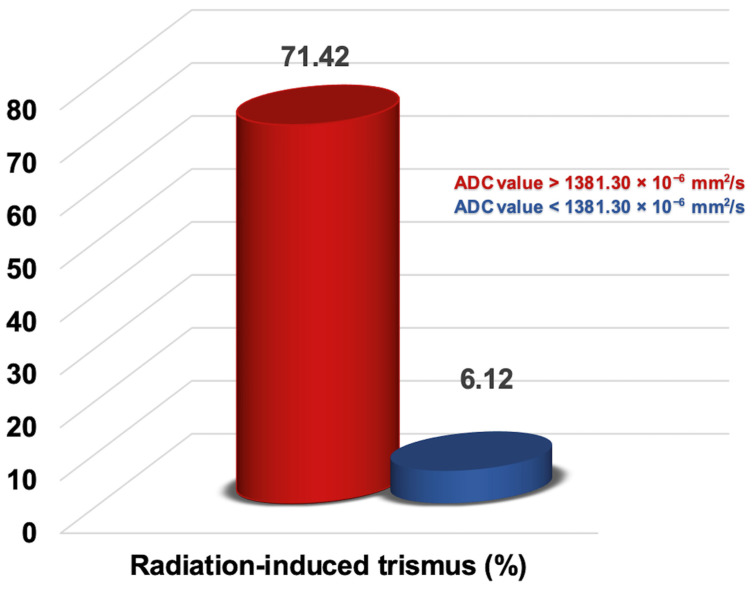
Representation of the radiation-induced trismus rates in the two groups categorized by the apparent diffusion coefficient (ADC) value of the masseter muscles.

**Figure 6 diagnostics-14-02268-f006:**
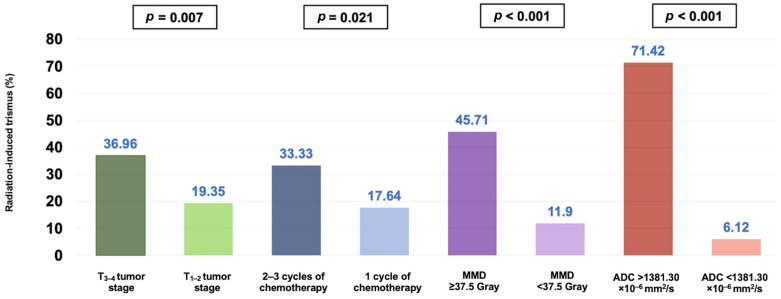
Bar graph representation of the ratio of radiation-induced trismus for the entire cohort, categorized by tumor stage, number of chemotherapy cycles, radiation dose to the masseter muscle, and pretreatment mean apparent diffusion coefficient (ADC) of the masseter muscles (MMD: masseter muscle dose).

**Table 1 diagnostics-14-02268-t001:** Baseline clinical, radiological, and treatment characteristics of locally advanced nasopharyngeal cancer patients and average ADC_mean_ of masseter muscle groups.

Characteristics	All Patients (*n* = 77)	Average ADC_mean_ of MMs < 1381.30 × 10^−6^ mm^2^/s (n = 49)	Average ADC_mean_ of MMs > 1381.30 × 10^−6^ mm^2^/s (n = 28)	*p*-Value
Median age, year (range)	53 (25–76)	53 (30–76)	54 (25–75)	
Age group, n (%)				
≥65 years	13 (16.88)	8 (16.33)	5 (17.86)	0.42
<65 years	64 (83.12)	41 (83.67)	23 (82.14)	
Gender, n (%)				
Female	22 (28.57)	15 (26.32)	7 (25)	0.41
Male	55 (71.43)	34 (73.68)	21 (75)	
ECOG, n (%)				
0	35 (45.45)	23 (46.94)	13 (46.43)	0.92
1	42 (54.55)	26 (53.06)	15 (53.57)	
T-stage *, n (%)				
1–2	31 (40.26)	19 (38.78)	12 (42.86)	0.44
3–4	46 (59.74)	30 (61.22)	16 (57.14)	
N-stage *, n (%)				
0–1	26 (33.77)	17 (34.69)	9 (32.14)	0.58
2–3	51 (66.23)	32 (65.31)	19 (67.86)	
Concurrent chemotherapy cycles, n (%)				
1	17 (22.08)	11 (22.45)	6 (21.43)	0.62
2–3	60 (77.92)	38 (77.55)	22 (78.57)	
Adjuvant chemotherapy cycles, n (%)				
0	20 (25.97)	13 (26.53)	7 (25)	0.78
1–2	57 (74.03)	36 (73.47)	21 (75)	
Mean masseter muscle dose, Gy (range)	37.5 (8.6–62.3)	38.3 (8.6–60.7)	36.6 (9.3–62.3)	0.49
Mean masseter muscle dose group, n (%)				
<37.5 Gy	42 (54.55)	26 (53.1)	16 (57.1)	0.51
≥37.5 Gy	35 (45.45)	23 (46.9)	12 (42.9)	

* According to the eighth edition of the *AJCC*. ADC, apparent diffusion coefficient; MMs, masseter muscles; ECOG, Eastern Cooperative Oncology Group; T-stage, tumor stage; N-stage, node stage; Gy: gray.

**Table 2 diagnostics-14-02268-t002:** Outcomes of univariate and multivariate analyses.

Variable	All Patients (n = 77)	RIT (%) (n = 23)	Univariate *p*-Value	Multivariate *p*-Value	HR (95% CI)
Age group, n (%)					
≥65 years	13	6 (46.15)	0.14	-	-
<65 years	64	17 (26.56)			
Gender, n (%)					
Female	22	7 (31.82)	0.50	-	-
Male	55	16 (29.09)			
ECOG, n (%)					
0	35	10 (28.5)	0.82	-	-
1	42	13 (30.95)			
T-stage *, n (%)					
1–2	31	6 (19.35)	0.003	0.007	1.84 (1.26–3.18)
3–4	46	17 (36.96)			
N-stage *, n (%)					
0–1	26	7 (26.92)	0.32	-	-
2–3	51	16 (31.37)			
Concurrent chemotherapy cycles, n (%)					
1	17	3 (17.64)	0.012	0.021	1.67 (1.21–2.06)
2–3	60	20 (33.33)			
Adjuvant chemotherapy cycles, n (%)					
0	20	5 (25.00)	0.37	-	-
1–2	57	18 (31.58)			
Mean masseter muscle dose group, n (%)					
<37.5 Gy	42	5 (11.90)	<0.001	<0.001	4.89 (2.66–7.93)
≥37.5 Gy	35	16 (45.71)			
ADC group, n (%)					
<1381.30 × 10^−6^ mm^2^/s	49	3 (6.12)	<0.001	<0.001	11.17 (5.63–18.76)
>1381.30 × 10^−6^ mm^2^/s	28	20 (71.42)			

* According to the eighth edition of the *AJCC*. RIT, radiation-induced trismus; HR, hazard ratio; CI, confidence interval; ECOG, Eastern Cooperative Oncology Group; T-stage, tumor stage; N-stage, node stage; Gy: gray.

## Data Availability

The present data belong to and are stored at Baskent University’s Faculty of Medicine and cannot be shared without permission. For the researchers who meet the criteria for access to confidential data, contact Baskent University’s Corporate Data Access/Ethics Board: adanabaskent@baskent.edu.tr.

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
