# Peer review of "The Prediction of Radiation-Induced Trismus by the Apparent Diffusion Coefficient Values of Masseter Muscles before Chemoradiotherapy in Locally Advanced Nasopharyngeal Carcinomas"

_diagnostics, 2024, doi:10.3390/diagnostics14202268_

Round 1
Reviewer 1 Report
Comments and Suggestions for Authors
The aim of the study titled "The Prediction of the Radiation–Induced Trismus by Apparent Diffusion Coefficient Values of Masseter Muscles Before Chemoradiotherapy in Locally Advanced Nasopharyngeal Carcinomas" is to investigate the significance of pre-concurrent chemoradiotherapy (CCRT) mean apparent diffusion coefficient (ADC) values of the masseter muscles in predicting the rates of radiation-induced trismus (RIT) in patients with locally advanced nasopharyngeal carcinoma (LA-NPC) undergoing definitive CCRT. The study seeks to establish a correlation between the ADC values and the likelihood of developing trismus as a side effect of the treatment, thereby potentially providing a biomarker for predicting patient outcomes.
Comments:
1. Introduction:
· The introduction effectively outlines the background and significance of the study. However, it could benefit from a more detailed discussion of existing literature on ADC values and their implications in oncology.
2. Materials and Methods:
· The methodology is generally well-described. The use of receiver operating characteristic curve analysis to determine the optimal ADC mean cutoff is appropriate. However, please clarify the sample size calculation and justify the number of patients included in the study.
· It would be helpful to include more details on the imaging techniques used, including the specific MRI protocols and any potential limitations related to the imaging equipment.
3. Results:
· Consider providing additional visual aids, such as graphs or tables, to enhance the presentation of key findings.
4. Discussion:
· The discussion effectively contextualizes the findings within the existing literature. However, it would be beneficial to explore the biological mechanisms that might explain the observed relationship between ADC values and RIT.
· The limitations section should be expanded to address potential confounding factors and the generalizability of the findings.
5. Conclusions:
· The conclusions are well-supported by the data. It would be advantageous to suggest future research directions or clinical applications based on the findings.
6. Minor Comments:
· Please ensure that all references are formatted consistently according to the journal's guidelines.
· There are minor grammatical errors and typos throughout the manuscript that should be corrected for clarity.
Author Response
We want to express our gratitude to the reviewer for his/her thoughtful recommendations. We have endeavored to address the inquiries and incorporate relevant changes in the text in accordance with the suggestions.
Comments:
- Introduction:
- The introduction effectively outlines the background and significance of the study. However, it could benefit from a more detailed discussion of existing literature on ADC values and their implications in oncology.
Response:
As recommended, we have added further data to the revised Introduction section concerning the implications of ADC values in oncological practice.
- Materials and Methods:
- The methodology is generally well-described. The use of receiver operating characteristic curve analysis to determine the optimal ADC mean cutoff is appropriate. However, please clarify the sample size calculation and justify the number of patients included in the study.
Response:
As this study represents the inaugural investigation into this particular subject matter, no pertinent data regarding the potential for ADC values to discriminate between RIT rates were available. Consequently, due to the absence of prior data or evidence-based anticipations, calculating a sample size was unfeasible, necessitating the inclusion of all eligible patients in the study.
- It would be helpful to include more details on the imaging techniques used, including the specific MRI protocols and any potential limitations related to the imaging equipment.
Response:
As recommended, the revised Imaging Acquisition, Post-processing Image Analysis, and Limitations sections detail these issues.
- Results:
- Consider providing additional visual aids, such as graphs or tables, to enhance the presentation of key findings.
Response:
In accordance with your valuable recommendations, a representative patient with RIT has been included to illustrate patient evaluation in the study. Furthermore, an additional graph has been incorporated to elucidate the disparity in RIT rates between the two ADC groups.
- Discussion:
- The discussion effectively contextualizes the findings within the existing literature. However, it would be beneficial to explore the biological mechanisms that might explain the observed relationship between ADC values and RIT.
Response:
As recommended, the potential biological mechanisms beyond a high pretreatment ADC value and higher RIT rates are detailed in a separate paragraph before the revised manuscript's Limitations section.
- The limitations section should be expanded to address potential confounding factors and the generalizability of the findings.
Response:
Following your recommendations, we have expanded the limitation section to address potential confounding factors and provide a more thorough examination of the findings' generalizability.
- Conclusions:
- The conclusions are well-supported by the data. It would be advantageous to suggest future research directions or clinical applications based on the findings.
Response:
In accordance with your recommendations, the Conclusions section has been revised as “In conclusion, the present study's findings suggest a strong and independent link between higher average ADCmean values in MMs of LA-NPC patients undergoing definitive CCRT and increased RIT incidence rates. If subsequent research corroborates these findings, assessing the association between pre-CCRT DWI findings and the risk of RIT could facilitate closer monitoring of high-risk patients and the timely implementation of preventive interventions. This strategy has the potential to significantly improve patients' quality of life.”
- Minor Comments:
- Please ensure that all references are formatted consistently according to the journal's guidelines.
Response:
All references have been meticulously reviewed and adjusted to adhere to the specifications outlined by the journal.
- There are minor grammatical errors and typos throughout the manuscript that should be corrected for clarity.
Response:
A team of professional commercial English editors thoroughly examined the revised manuscript to identify specific errors and implemented the recommended corrections.
Reviewer 2 Report
Comments and Suggestions for Authors
Pehlivan et al investigated the significance of pre-concurrent chemoradiotherapy masseter muscle apparent diffusion coefficient mean values in predicting the radiation-induced trismus rates in locally advanced nasopharyngeal carcinoma patients treated with definitive CCRT and obtained very interesting results. More precisely, the authors concluded that a higher pre-CCRT MM average ADCmean value (> 1381.30 × 10−6 mm2/s) is linked to significantly increased rates of RIT (71.43% vs. 6.12% for average ADCmean < 1381.30 × 10− 6 mm2/s; HR: 11.17, p < 0.001) in LA-NPC patients who received definitive CCRT. At the same time, they confirmed the traditional factors for the development of RIT, such as a higher stage of the disease, radiation dose to the masseter, concomitant chemotherapy, etc. The study is certainly commendable, but these are my questions:
1. Is it standard to perform an MRI on every patient with locally advanced nasopharyngeal carcinoma? Or is a CT scan sufficient?
2. The patients included in the study are from 2010 to 2023, the maximum opening of the mouth was measured at 1, 3, 6, 9 and 12 months? Is this a standard that you perform or was it done just for this study? In fact, the protocol described in this way is more suitable for prospective research, and the study itself is retrospective. If you could explain that part more precisely?
Regardless, the paper is clearly written, the discussion is interesting, the limitations of the study are precisely stated, as is the conclusion, which suggests more of a hypothetical. In any case, prospective studies on this topic are needed.
Author Response
We extend our gratitude to the reviewer for their thoughtful recommendations. We have diligently worked to address the inquiries and integrate pertinent changes into the text following the suggestions.
- Is it standard to perform an MRI on every patient with locally advanced nasopharyngeal carcinoma? Or is a CT scan sufficient?
Response:
MRI is essential for detecting early NPC, staging the primary tumor, and evaluating associated retropharyngeal and cervical lymphadenopathy. MRI can also detect submucosal lesions unseen in endoscopy. It has been used to monitor patients after therapy to detect tumor recurrence and radiation-associated soft tissue and bone changes. Imaging is valuable for the differentiation of NPC from other simulating lesions. Therefore, we use MRI as the standard staging, tumor volume delineation (co-registered with planning CT), and follow-up tool for all NPCs in our clinic, which are in line with the following guideline recommendations. The National Comprehensive Cancer Network (NCCN) guidelines for head and neck cancers include the following recommendations for imaging of NPC:
- Initial imaging: MRI with contrast is preferred over CT; imaging should include the skull base to the clavicle.
- Inconclusive initial imaging: PET/CT should be performed to identify primary sites before intervention.
- Distant metastases: FDG PET/CT and/or chest CT with contrast.
In 2021, the European Society for Medical Oncology and EURACAN released joint guidelines for the diagnosis, treatment, and follow-up of nasopharyngeal carcinoma. The recommendations concur with the NCCN guidelines. The European guidelines also include the following imaging recommendations for staging NPC:
- Local and nodal tumor staging: MRI is preferred; FDG-PET may be used to add further accuracy in nodal staging.
- Locally advanced disease or distant metastases staging: FDG-PET.
- The patients included in the study are from 2010 to 2023, the maximum opening of the mouth was measured at 1, 3, 6, 9 and 12 months? Is this a standard that you perform or was it done just for this study? In fact, the protocol described in this way is more suitable for prospective research, and the study itself is retrospective. If you could explain that part more precisely?
Response:
At our institution, head and neck cancer patients undergoing radiotherapy or chemoradiotherapy receive regular examinations by an experienced oral and maxillofacial surgeon specialized in dental oncology. These assessments take place prior to treatment, at 1, 3, 6, 9, and 12 months post-treatment, and subsequently at 6-month intervals.